# Faster Video Diffusion with Trainable Sparse Attention

**Peiyuan Zhang**[1*]    **Yongqi Chen**[1*]    **Haofeng Huang**[1*‡]    **Will Lin**[1]
**Zhengzhong Liu**[2]    **Ion Stoica**[3]    **Eric P. Xing**[2]    **Hao Zhang**[1]
[1]UC San Diego    [2]MBZUAI    [3]UC Berkeley

## Abstract

Scaling video diffusion transformers (DiTs) is limited by their quadratic 3D attention, even though most of the attention mass concentrates on a small subset of positions. We turn this observation into VSA, a trainable, hardware-efficient sparse attention that replaces full attention at *both* training and inference. In VSA, a lightweight coarse stage pools tokens into tiles and identifies high-weight *critical tokens*; a fine stage computes token-level attention only inside those tiles subjecting to block computing layout to ensure hard efficiency. This leads to a single differentiable kernel that trains end-to-end, requires no post-hoc profiling, and sustains 85% of FlashAttention3 MFU. We perform a large sweep of ablation studies and scaling-law experiments by pretraining DiTs from 60M to 1.4B parameters. VSA reaches a Pareto point that cuts training FLOPS by $2.53\times$ with no drop in diffusion loss. Retrofitting the open-source Wan2.1-1.3B model speeds up attention time by $6\times$ and lowers end-to-end generation time from 31s to 18s with comparable quality, while for the 14B model, end-to-end generation time is reduced from 1274s to 576s. Furthermore, we introduce a preliminary study of Sparse-Distill, the first method to enable sparse attention and distillation concurrently, achieving 50.9x speed up for Wan-1.3B while maintaining quality. These results establish trainable sparse attention as a practical alternative to full attention and a key enabler for further scaling of video diffusion models. Code is available at https://github.com/hao-ai-lab/FastVideo.

## 1   Introduction

Attention computation is the primary bottleneck when scaling video Diffusion Transformers (DiT) [34, 28]. Even a seemingly short 5-second 720p clip unfolds into more than 100K tokens [29, 20] once flattened as a sequence. Consequently, state-of-the-art video DiTs [20, 35, 43, 26] expend the majority of compute on attention when training on full-resolution, long-sequence data; the trained DiTs remain painfully slow at inference. Fortunately, recent studies [37, 48, 6, 47] reveal an inherent sparsity in DiTs trained using full attention: only a tiny subset of entries in the attention matrix $\text{Softmax}(QK^\top/\sqrt{d})$, which we refer to as *critical tokens*, significantly influence outputs, while the vast majority approach zero. This inherent sparsity calls for developing a native, trainable sparse attention mechanism purpose-built for video DiTs.

Most prior work approaches sparsity as a post-hoc speedup for pretrained DiTs rather than as a first-class training primitive. Sliding Tile Attention (STA) [48] and Sparge Attention [47], for example, begin with a model trained under full attention and then substitute each head with a fixed or profile-derived sparse mask only at inference time [37, 38]. Because the sparsity pattern is decided *after* training, these methods leave the bulk of training cost untouched and introduce a train-test mismatch: the DiT learns parameters in a dense context yet is evaluated in a sparse one. That mismatch caps

---

[*]Equal contribution. [‡]Work performed during an internship at UC San Diego.

best-case quality at the dense model's ceiling and, in practice, often erodes quality once sparsity is pushed beyond a gentle budget. Consequently, state-of-the-art DiTs still default to quadratic 3D attention despite its prohibitive cost [43, 35, 20, 2, 11].

Designing a trainable sparse attention for video DiTs faces a fundamental chicken-and-egg dilemma: identifying critical token positions traditionally requires computing the full attention matrix, which then erases any computational gains and defeats the purpose of sparse attention. Conversely, resorting to cheap heuristics and not precisely identifying critical tokens may miss high-weight regions and yield suboptimal results. More importantly, any practical attention implementation must honor the block-sparse layouts expected by modern GPU kernels such as Flash Attention (FA) [5] – otherwise theoretical savings do not translate to wall-clock speedup. The central research question is therefore: how can we predict critical tokens accurately, subject to hardware-aligned block structure, without paying the quadratic cost we aim to avoid?

This paper presents VSA (Video Sparse Attention), a trainable, hardware-aligned sparse attention mechanism for video DiTs, drawing inspiration from recent developments in large language models [45, 25, 30]. At the high level, VSA is a hierarchical granular attention, illustrated in Figure 1. The coarse stage first aggregates a cube containing $(4, 4, 4)$ tokens into a single representation to compute cube-to-cube dense attention. Since attention operates on a pooled (short) sequence, it is lightweight, yet *simultaneously* predicts which cube contains critical tokens while modeling global context. A fine stage then applies token-level attention *only* inside the top-$\mathcal{K}$ selected cube. The final output of attention combines both stages through a differentiable gate. Because VSA is end-to-end trainable, it identifies critical tokens not by heuristics, but by *learning from data*. To ensure hardware efficiency, VSA is meticulously designed to map a spatial-temporal cube to a kernel-level tile [1] [48]. This ensures that tokens within a cube are loaded on the same GPU SM and adhere to the block sparse compute layout (§2.2).

One critical parameter in VSA is the tile size. Small tiles let the coarse stage localize critical tokens close to token resolution, while the fine stage attends only to those pinpointed cubes. This sharpens sparsity and improves model quality at the cost of fragmenting work into tiny blocks, which hurts the GPU throughput. In contrast, large tiles boost arithmetic intensity, but the fine stage must then process whole cubes even if only a few tokens inside the cubes matter, thus blurring sparsity (Figure 1 (b)). Another key parameter is whether to inject dedicated local or spatial-temporal patterns into the fine stage. Our systematic ablation studies reveal that an effective VSA configuration combines a global coarse stage with a freely-selectable fine stage. We found that a tile size of 64 and 87.5% attention sparsity achieves performance comparable to full attention while maintaining efficient kernel execution. Furthermore, purposely injecting locality heuristics proved unnecessary. A large sweep of scaling experiments, in which we pretrain video DiTs from scratch (60M to 1.4B parameters with up to $4 \times 10^{21}$ FLOPS) with 16K sequence length, uncovers a Pareto frontier where VSA achieves near $8\times$ reduction in attention FLOPS and $2.53\times$ reduction in total training FLOPS.

To support VSA's hierarchical sparse attention, we prototype GPU kernel where the coarse stage kernel fuses softmax, Top-$\mathcal{K}$ selection, and block indexing into a single pass, and the fine stage leverages a block-sparse attention kernel based on FA [5]. This leads to our VSA implementation that retains $85\%$ of FA3's MFU [31]. We further retrofit VSA into SoTA open source DiT, Wan2.1 1.3B [35], originally trained with full attention. This integration speeds up attention time by 6x and reduces end-to-end inference latency from 31s to 18s (1.7x) on H100. As a result, VSA enables the first video DiT where attention accounts for only 20% of runtime during both training and inference without quality degradation.

To the best of our knowledge, VSA is the first trainable sparse attention approach that, based on extensive experiments totaling around 90k H200 hours, shows better scaling than full attention on DiTs. Finally, we hope that our explicit ablation of the key parameters (e.g., tile size, critical token prediction, locality prior, sparsity, etc.) will enable more targeted exploration of sparse attention in scaling video DiTs.

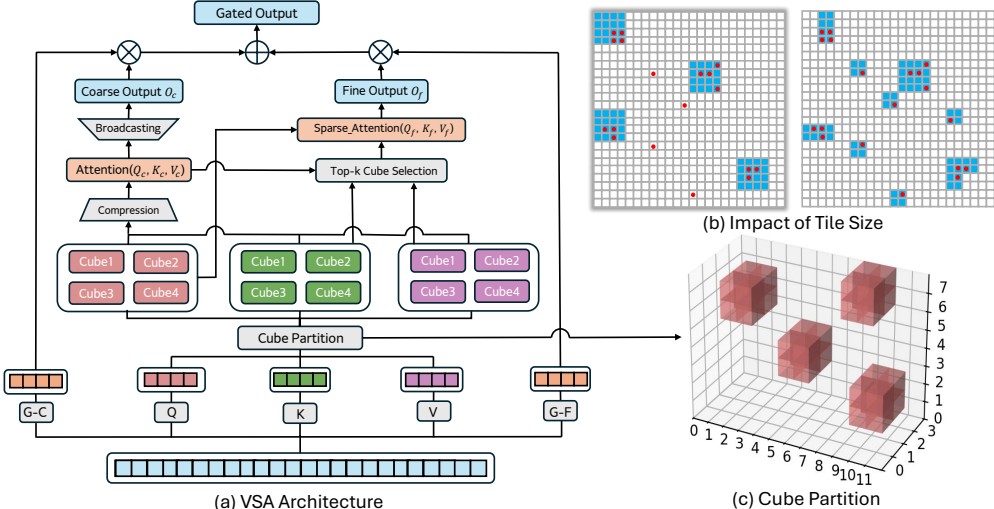

(b) Impact of Tile Size

(a) VSA Architecture

(c) Cube Partition

Figure 1: Overview of VSA. (a) VSA introduce a hierarchical attention with sparsity and different granularity (coarse & fine). (b) Larger tile sizes (left) blur attention pattern while small tiles let the coarse stage localize critical tokens close to token resolution. The red dots indicate critical tokens. (c) An illustration of $(2, 2, 2)$ cube partition (in practice we use $(4, 4, 4)$ in VSA).

## 2 Methods

This section introduces VSA, our sparse attention designed to reduce both training and inference costs of video DiTs. We begin by detailing the design space, key components, and core motivations behind sparse attention in § 2.1. Although the final method is presented in § 2.2, we emphasize that design exploration and ablation studies (deferred to § 3) are central to understanding the effectiveness of VSA. Finally, § 2.3 describes how to adapt VSA to pretrained Video DiTs originally trained with full attention.

### 2.1 Sparse Attention Design Space

Modern video DiTs use 3D full attention to capture dependencies across the entire video volume. Given a video latent of shape $(T, H, W)$, it is flattened into a 1D sequence of length $L = THW$ by mapping each token location $(t, w, h)$ in the 3D latent to its position $n$ in the 1D sequence following $n = tHW + hW + w$. Then full attention is applied across the entire 1D sequence, allowing each token to interact with all the others. Let $\mathbf{Q}, \mathbf{K}, \mathbf{V} \in \mathbb{R}^{L \times d}$ denote the query, key, and value matrices for a single attention head; let $\mathbf{M} \in \{-\infty, 0\}^{L \times L}$ denote the attention mask specifying the allowed connections between each pair of tokens. The attention output $\mathbf{O}$ is then calculated as:

$$\mathbf{S} = \frac{\mathbf{Q}\mathbf{K}^\top}{\sqrt{d_k}}, \quad \mathbf{A} = \text{Softmax}(\mathbf{S} + \mathbf{M}), \quad \mathbf{O} = \mathbf{A}\mathbf{V}$$

**Block Size vs. Hardware Efficiency.** In *full attention*, all entries in $\mathbf{M}$ are zero. *Sparse attention* introduces $-\infty$ entries in $\mathbf{M}$ and theoretically reduces the total FLOPS by allowing the computation of the corresponding elements in both $\mathbf{Q}\mathbf{K}^\top$ and $\mathbf{A}\mathbf{V}$ to be skipped. However, modern accelerators are optimized for dense computation, making unstructured sparsity ineffective for real speedup. *Block-sparse attention* [5] addresses this by structuring the sparsity to align with the hardware capabilities. In this approach, the attention mask $\mathbf{M}$ is divided into tiles of size $(B_q, B_k)$ [2], with all entries of

---

[1]We use the word *block* and *tile* interchangeably in this paper. They refer to a submatrix that a GPU threadblock loads into SRAM when performing matrix multiplication.

[2]Technically $(B_q, B_k)$ can be non-square with different values for $B_q$ and $B_k$. To keep the notation simple, we assume that $B = B_q = B_k$.

each tile sharing the same value. This enables each tile in a GPU SM to be processed as a dense block or skipped entirely, maximizing hardware efficiency. The tile size $B$ is a key design parameter: smaller tiles allow flexible, fine-grained sparsity but are less efficient on hardware, while larger tiles improve throughput but restrict the model to coarser attention patterns, potentially reducing modeling expressiveness (see Figure 1 (b)). Thus, selecting $B$ involves a tradeoff between expressiveness and efficiency. In practice, we find that modest reductions in speed can be acceptable if they yield significant improvements in generation quality.

**Prediction Cost vs. Coverage Quality in Critical Token Selection.** Recent studies have shown that the attention score matrix $\mathbf{A}$ is naturally sparse [37, 48, 6, 47], with most values close to zero. This suggests that, by constructing a mask $\mathbf{M}$ that preserves only the high-value regions of $\mathbf{A}$ – the so-called critical tokens – we can closely approximate full attention while significantly reducing computation. A central design choice is how much computation to spend identifying these critical tokens. Computing full attention scores provides the most accurate selection, but largely negates computational savings, as only the $\mathbf{AV}$ operation benefits from sparsity. In contrast, fixed patterns (e.g., windowed or spatiotemporal attention) incur no prediction cost but often miss informative tokens. Inspired by NSA [45] and MoBA [25], we propose a lightweight, trainable, coarse-granular attention module to estimate the locations of critical tokens without fully computing $\mathbf{A}$. The main challenge is to balance prediction accuracy with computational efficiency for practical use in DiT architectures.

**Maintaining Global and Local Context in Sparse Attention.** A key challenge with sparse attention is its restricted receptive field, which can limit the model's ability to capture global context. One approach to address this is to augment sparse attention with a lightweight global module that captures coarse global signals. Conversely, incorporating local context—motivated by the locality priors commonly used in vision models such as CNNs—can also poteantially enhance feature learning. We empirically ablate both strategies to assess their impact on video generation quality in § 3.1.

## 2.2   VSA: Video Sparse Attention

VSA employs a cube-based partitioning strategy followed by a two-stage attention mechanism to efficiently process video latents. The method first divides the input video latent into spatially and temporally contiguous cubes, then processes these cubes through a coarse stage that identifies important regions and a fine stage that performs detailed token-level attention within these regions. This design enables efficient computation while maintaining the ability to capture both global and local dependencies in the video data.

Given a video latent with shape $(T, H, W)$, VSA divides it into multiple cubes, each with shape $(C_t, C_h, C_w)$ (Figure 1 (c)). VSA co-designs the sparse attention algorithm and its kernel implementation by mapping each cube in the video latent into a single tile on GPU SM, where the tile size $B = C_t \times C_h \times C_w$. We assume the video latent shape $(T, H, W)$ is an integer multiple of the tile size and define $(N_t, N_h, N_w) = \left( \frac{T}{C_t}, \frac{H}{C_h}, \frac{W}{C_w} \right)$. When flattening the 3D video latent into a 1D sequence, each token at position $(t, h, w)$ is assigned a 1D index $n$ using the following mapping:

$$ n = \left( \left\lfloor \frac{t}{C_t} \right\rfloor N_h N_w + \left\lfloor \frac{h}{C_h} \right\rfloor N_w + \left\lfloor \frac{w}{C_w} \right\rfloor \right) B + (t \bmod C_t) C_h C_w + (h \bmod C_h) C_w + (w \bmod C_w). $$

Building on this cube-based partitioning, VSA implements its two-stage attention mechanism to efficiently predict critical token locations *without* computing the full attention matrix $\mathbf{A}$, as shown in Figure 1 (a). In the *coarse stage*, we apply mean pooling over each $(C_t, C_h, C_w)$ cube to obtain cube-level representations, producing $\mathbf{Q_c}, \mathbf{K_c}, \mathbf{V_c} \in \mathbb{R}^{\frac{L}{B} \times d}$. This stage then computes attention scores $\mathbf{A_c} \in \mathbb{R}^{\frac{L}{B} \times \frac{L}{B}}$ and outputs $\mathbf{O_c}$. The attention mask $\mathbf{M}$ is derived by selecting the Top-$\mathcal{K}$ entries per row in $\mathbf{A_c}$ and setting others to $-\infty$, followed by broadcasting to a full-resolution mask of size $L \times L$. This mask naturally conforms to a block-sparse structure because the coarse stage operates on cube-level representations. When broadcasting the mask from $\frac{L}{B} \times \frac{L}{B}$ to $L \times L$, each selected entry in $\mathbf{A_c}$ expands into a $B \times B$ block in $\mathbf{M}$[3]. This block-sparse pattern is crucial for hardware efficiency as it allows the next stage to process attention in contiguous blocks that align with GPU memory access patterns and enable efficient parallel computation.

---

[3]In practice we do not broadcast to full-resolution mask but only input the selected block index to fine stage.

Next, in the *fine stage*, this mask $\mathbf{M}$ guides fine-grained attention computation for $\mathbf{Q}, \mathbf{K}, \mathbf{V} \in \mathbb{R}^{L \times d}$, yielding output $\mathbf{O_f}$. Finally, outputs from both stages are combined to obtain the final output $\mathbf{O}$:

$$\mathbf{O} = \mathbf{O_c} \odot \mathbf{G_c} + \mathbf{O_f} \odot \mathbf{G_f}$$

where $\mathbf{G_c}$ and $\mathbf{G_f}$ are gating vectors obtained from linear projections of the input hidden states. Since the coarse stage introduces negligible computational cost (less than 1% of total FLOPS), the overall sparsity can be approximated by $\frac{\mathcal{K}B}{L}$. However, due to the row-wise Top-$\mathcal{K}$ selection, FlashAttention cannot be directly applied to the coarse stage, resulting in increased latency. § 2.4 discusses how we mitigate this overhead. Appendix B gives a pseudo code implementation of VSA .

In § 3.1 and 3.2, we show that using a smaller $B$ leads to a more expressive sparse attention pattern and improved performance, but comes at the cost of slower attention kernel execution. Setting $B = 64$ with $(C_t, C_h, C_w) = (4, 4, 4)$ provides a favorable trade-off between expressiveness and efficiency. We further demonstrate that combining a coarse stage for global context with a fine stage for token-level sparse attention is both necessary and sufficient—dedicated modules for local context modeling offer minimal additional benefit. Setting $\mathcal{K} = 32$ consistently yields strong performance across a wide range of sequence lengths. VSA adopts these hyperparameters as default.

## 2.3 Sparse Adaptation & Distillation

VSA is designed to train video DiTs from scratch, reducing both training and inference FLOPS. It can also be adapted to pretrained video DiTs originally trained with full attention. However, directly replacing full attention with VSA leads to unstable training. We hypothesize two main reasons: (1) the gating projection weights $\mathbf{G}$ are not present in the full attention checkpoint and are randomly initialized, and (2) VSA differs significantly from full attention, introducing a coarse stage and sparsity in the fine stage. To address this, we develop an annealing strategy that smoothly transitions the model from full attention to VSA. We initialize the weights of the coarse gate $\mathbf{G_c}$ to zero, and remove the fine gate $\mathbf{G_f}$ (equivalent to $\mathbf{G_f} = \mathbf{1}$). We also initialize the sparsity level by setting $\mathcal{K} = \frac{B}{L}$, effectively making VSA equivalent to full attention at the start of training. As training progresses, we gradually reduce $\mathcal{K}$ to the target sparsity level. Meanwhile, $\mathbf{G_c}$ is updated through training, enabling the model to learn how to balance the contributions of the coarse- and fine stages.

In § 3.3, we demonstrate that Wan-2.1 can be efficiently converted into a sparse-attention model by finetuning with a standard flow-matching loss for only a small number of steps. We further conduct a preliminary study on integrating sparse attention with distillation [44]. In this setup, the student is trained to act simultaneously as a few-step generator and a sparse generator, while the teacher remains unchanged with full attention. Importantly, we preserve the original distillation loss and all hyperparameters, requiring no modifications beyond adapting the student. To our knowledge, VSA is the first sparse attention method shown to be compatible with distillation, whereas prior approaches that exploit diffusion time-step redundancy may fail in the extremely low-step distillation regime.

## 2.4 Kernel Implementation

VSA requires implementing both forward and backward kernels. We write block-sparse attention kernel with ThunderKittens [32] for fine stage. Despite using a relatively small tile size of 64, our kernel achieves over 85% of FA's MFU (§3.4). The coarse stage, illustrated in Figure 1, requires row-wise Top-$\mathcal{K}$ selection over the cube-level attention matrix. This step necessitates materializing the attention matrix, which precludes direct use of FA-style fused kernels.

One possible workaround is to modify the FA kernel to incorporate in-kernel bitonic sorting for Top-$\mathcal{K}$, thereby avoiding materialization. However, such fusion demands intrusive kernel rewriting and careful tuning. We instead ask: is this complexity necessary? For VSA, the coarse stage operates on $(4, 4, 4)$ cubes, reducing sequence length by $64\times$, e.g., a 100K-token sequence reduces to 1.5K. At this scale, the memory overhead from materialization is negligible. FLOPS-wise, the coarse stage contributes less than 0.2% of total attention compute, and we show in §3.4 that its runtime accounts for only 14%, even when the fine stage is 87.5% sparse. This makes further kernel fusion unnecessary. Nonetheless, we still make some efforts to speed up the coarse stage. Our block-sparse kernel consumes block indices, not binary masks. Therefore, converting the Top-$\mathcal{K}$ mask into index form incurs additional overhead. To mitigate this, we fuse softmax, Top-$\mathcal{K}$ selection, and mask-to-index conversion into a single kernel. This fused kernel reduces coarse stage runtime modestly(§3.4D).

| Exp ID | Case | Loss – Opt | Loss – Over |
|---|---|---|---|
| 1 | Compress KV | 0.15281 | 0.14282 |
| 2 | Spatial Temporal | 0.13574 | 0.13034 |
| 3 | Spatial Full | 0.13555 | 0.12811 |
| 4 | Strided Window | 0.13271 | 0.12716 |
| 5 | Full | 0.13877 | 0.12703 |
| 6 | VSA | **0.13162** | **0.12687** |

(a) VSA v.s. other attention.

| Exp ID | Case | Loss |
|---|---|---|
| 7 | L | 0.13330 |
| 8 | F (no $\mathbf{O_c}$) | 0.13296 |
| 9 | C & L | 0.13220 |
| 10 | C & F | 0.13162 |
| 11 | C & F & L | 0.13194 |
| 12 | C & F & L & E | **0.13124** |
| 13 | C & (F + L) | 0.13192 |

(b) Different attention design.

| $B$ | $(C_t, C_h, C_w)$ |
|---|---|
| 256 | (4, 8, 8) |
| 128 | (4, 8, 4) |
| 64 | (4, 4, 4) |
| 16 | (2, 4, 2) |

(c) $B$ to $(C_t, C_h, C_w)$.

| Exp ID | C Pooling | F Tile | TFLOPS | Loss |
|---|---|---|---|---|
| 14 | 256x256 | 256x256 | 478 | 0.13375 |
| 15 | 128x128 | 128x128 | 444 | 0.13244 |
| 16 | 64x64 | 256x256 | 478 | 0.13328 |
| 17 | 64x64 | 64x64 | 408 | 0.13162 |
| 18 | 16x64 | 16x64 | 181 | **0.13155** |

(d) Tile size ablation.

| Exp ID | Pooling | Loss |
|---|---|---|
| 19 | Conv | 0.27787 |
| 20 | Max | 0.13929 |
| 21 | Avg | **0.13162** |

(e) Critical token prediction study.

Table 1: Ablation results for key design parameters of VSA.

## 3 Experiments

### 3.1 Ablation Studies

We conduct extensive pretraining experiments to ablate various design choices. Our experiments are based on the Wan2.1 model architecture, a state-of-the-art open-source video DiT. Unless otherwise specified, we train models with 120M parameters from scratch for $4.5 \times 10^{20}$ FLOPS using video latents of shape $(16, 32, 32)$ from the Vchitect-T2V-Dataverse dataset [10]. We determined $4.5 \times 10^{20}$ to be compute-optimal for our setup by following established scaling laws [15, 19]: when comparing models of different sizes under fixed compute budgets, we observe that a 120M parameter model with full attention trained at $4.5 \times 10^{20}$ FLOPS achieves better performance than smaller models (60M) with the same compute budget, while increasing compute beyond this point yields diminishing returns. For VSA and its variants, we set the number of selected KV tiles $\mathcal{K}$ to 32, achieving an attention sparsity of approximately 87.5%. Detailed experimental setup and the rationale behind our experiment design can be found in Appendix C. Key findings from these ablations are presented below.

**Data-Dependent Trainable Sparsity Wins Over Fixed-Patterns.** We first investigate why full attention predominates despite sparse alternatives. Table 1a shows existing sparse methods (Exp 1-4) outperform full attention (Exp 5) with a compute-optimal training budget $(4.5 \times 10^{20}$ FLOPS), but this advantage reverses with extended training $(4 \times 10^{21}$ FLOPS). VSA (Exp 6) outperforms both previous fixed-pattern methods and full attention. To understand why, in Table 1b(b) we examine two key factors: pattern type and stage contributions. We compare data-dependent patterns against fixed local patterns ("L") that use a $(3, 3, 3)$ window. Simultaneously, we investigate the impact of the coarse stage by including or excluding its output $\mathbf{O_c}$ (denoted as "C") from the final attention output, versus using only the fine stage output (denoted as "F"). Data-dependent patterns consistently outperform fixed patterns, both without the gated residual (Exp 7 vs. 8) and with it (Exp 9 vs. 10), demonstrating the inherent advantage of adaptive sparsity and the gated residual.

**Global Information is Necessary; Locality Priors Offer Limited Gains.** We tested three approaches for incorporating local context: (1) adding a separate local stage for $(3, 3, 3)$ window attention (Exp 11), (2) explicitly excluding ("E") cubes selected by the local stage from the fine stage (Exp 12), and (3) forcing the fine stage to include local cubes, without a separate local stage (Exp 13). All three variations performed similarly to the simpler C & F architecture, indicating that explicit local modeling provides minimal benefit. VSA therefore adopts the simpler C & F architecture (Exp 10), which effectively captures both global and local information.

**Finegrained Attention Map Leads to Better Performance But a Slower Kernel.** As analyzed in Section 2.1, the tile size $B$ in VSA is a critical hyperparameter that balances computational efficiency against model performance. It directly affects two key aspects: (1) how accurately critical tokens can be identified through the coarse stage's cube size, and (2) the granularity of attention in the fine stage. Hardware constraints partially dictate this parameter—NVIDIA Hopper GPUs optimize for

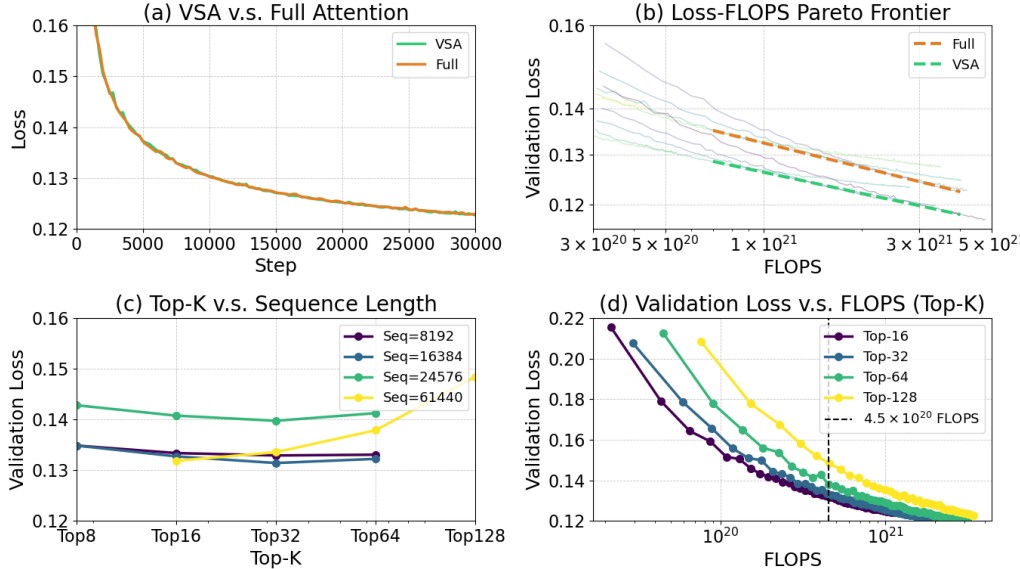

Figure 2: VSA scaling experiments. (a): Video DiT trained with VSA achieves similar loss curve compared to one trained with full attention. (b): VSA consistently produces a better Pareto frontier when scaling model size up to 1.4B. (c) & (d): The optimal Top-$\mathcal{K}$ value (dictating sparsity) depends on both sequence length and training compute. A larger $\mathcal{K}$ is needed for a larger training budget.

matrix multiplications with dimensions divisible by 16, and smaller tiles generally reduce arithmetic intensity. Our benchmarks in Table 1d show that decreasing tile size from $256 \times 256$ to $64 \times 16$ significantly reduces Model FLOPS Utilization (MFU).

Training with various tile sizes (Table 1c) reveals smaller tiles consistently reduce model loss through finer attention granularity. This improvement stems from the finer granularity allowing the coarse stage to more accurately predict critical-token cubes and the fine stage to focus attention on smaller, more relevant regions. Experiment 16 specifically tests mismatched granularity between stages, confirming that both coarse and fine stage granularity significantly impact performance. Here, the coarse stage used smaller pooling cubes $(C_t, C_h, C_w) = (4, 4, 4)$ (effectively $B = 64$) while the fine stage operated on larger tiles corresponding to $(C_t, C_h, C_w) = (4, 8, 8)$ (effectively $B = 256$). To reconcile the finer granularity of the coarse stage's predicted attention map with the coarser block-sparse attention in the fine stage, we applied an additional $(1, 2, 2)$ average pooling before selecting top-$\mathcal{K}$ entries.

Balancing these findings, we selected $64 \times 64$ tiles $((C_t, C_h, C_w) = (4, 4, 4))$ as our default configuration. While $64 \times 16$ tiles offer slightly better performance, they run $2.26\times$ slower (Exp 18 vs. 17), making this tradeoff unfavorable.

**Mean Pooling Is Sufficient.** We also examined different pooling methods for the coarse stage. Table 1e shows that average pooling outperforms both max pooling and convolutional approaches, with the latter causing training instability.

### 3.2 Scaling Studies

To validate VSA, we pretrained a 410M video DiT with latent shape $(16, 32, 32)$ (16,384 tokens), larger than our 120M ablation models. Figure 2(a) shows VSA achieves nearly identical loss to full attention despite 87.5% sparsity ($\mathcal{K} = 32$ out of 256 cubes), while reducing attention FLOPS by $8\times$ and end-to-end training FLOPS by $2.53\times$. Further scaling experiments from 60M to 1.4B parameters (Figure 2(b)) confirm that VSA consistently produces a better Pareto frontier than full attention. The parallel fitted curves indicate that VSA maintains its $2.53\times$ FLOPS reduction across scales. Each model was trained with compute budgets up to $4 \times 10^{21}$ FLOPS on 128 H200 GPUs with sequence length of 16K. To our knowledge, VSA is the first trainable sparse attention for video DiTs demonstrating superior performance compared to full attention under rigorous scaling

| Model | Qual. | Sem. | Total |
|-------|-------|------|-------|
| Ori-Wan | 83.71% | 77.98% | 82.56% |
| Full f.t. | 84.07% | 81.85% | 83.63% |
| VSA f.t. | 83.60% | 79.47% | 82.77% |

(a) VSA Wan-1.3B results on VBench. With sparse adaptation, VSA is able to achieve similar score to a full attention counterpart and even slightly outperform the original model.

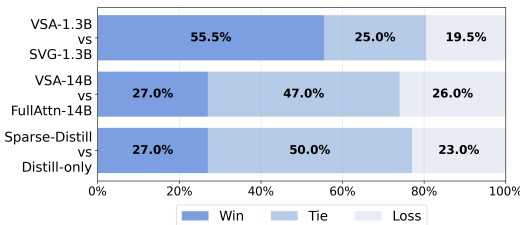

(b) Top: VSA vs. SVG human evaluation. SVG has a 82.5% attention sparsity with (fp0.03, fl0.025, s0.1). Middle: VSA v.s. full attention with finetuning. Bottom: VSA v.s. full attention with distillation.

evaluation. While we leave comprehensive scaling studies at longer sequence lengths to future work, our fine-tuned Wan 2.1 already shows $1.7\times$ inference speedup at 23K sequence length (§3.3).

An important design question for VSA is determining the optimal sparsity level via the Top-$\mathcal{K}$ parameter. In Figure 2(c), we pretrained 120M models with varying sequence lengths under a fixed $4.5 \times 10^{20}$ FLOPS budget. Surprisingly, $\mathcal{K} = 32$ performs consistently well across sequence lengths of 8192, 16384, and 24675, but underperforms compared to $\mathcal{K} = 16$ at 61440 sequence length. This contradicts the conventional intuition that longer sequences require higher $\mathcal{K}$ values. Further investigation with increased compute budget at 61440 sequence length (Figure 2(c)) reveals that $\mathcal{K} = 32$ eventually outperforms $\mathcal{K} = 16$ at $1 \times 10^{21}$ FLOPS, with similar patterns at other lengths. These findings suggest that optimal $\mathcal{K}$ depends on both sequence length and training budget. We hypothesize that the ideal Top-$\mathcal{K}$ increases with available compute, converging to full attention with infinite resources. However, precisely predicting optimal $\mathcal{K}$ given budget, model size, and sequence length remains an open question. One promising direction is to explicitly model sparsity as an additional axis in the scaling law framework [19, 15], alongside model size and total FLOPS. Incorporating inference costs further complicates this analysis, as higher $\mathcal{K}$ values may improve training loss but increase inference overhead. We leave a comprehensive treatment of these tradeoffs to future work.

### 3.3 Sparse Adaptation & Distillation

To evaluate VSA's effectiveness in a post-training setup, we finetune Wan2.1-1.3B with VSA on synthetic data generated by Wan-14B with video latent $16 \times 28 \times 52$ (480P). We set $\mathcal{K}$ to 32, corresponding to a 91.2% attention sparsity. As shown in Table 3a, VSA achieves even higher VBench [16] score compared to the original Wan-1.3B. We hypothesize training with synthetic data from a larger model may contribute to this boost. To ensure a fair comparison, we also finetune Wan-1.3B using the same synthetic data. The results show that all models perform closely on VBench, indicating that VSA can retain generation quality despite significant attention sparsity. We additionally compared VSA to SVG [37], a training-free attention sparsification method, under extreme sparsity. Figure 3b Top shows that VSA is preferred even though it has a higher sparsity, demonstrating the effectiveness of training with sparse attention. With VSA, the DiT inference time of Wan-1.3B drops from 31s (full attention with torch compile) to 18s.

To further validate the effectiveness of VSA across different model size and resolution, we scale our study to the Wan-14B model, finetuning on 720P synthetic data (latent $20 \times 48 \times 80$) at 90% sparsity. For this setup, we conduct human preference study on 200 randomly sampled MovieGen prompts [29] to compliment our 1.3B VBench results. As shown in Figure 3b Middle, human evaluation demonstrates that VSA preserves generation quality compared to the official full attention model, indicating that VSA is capable of maintaining high-quality performance at larger model scales. As a pilot study, we further explore whether sparse attention can complement other acceleration techniques, particularly distillation. In this setup, the student model in DMD2 [44] employs VSA while the teacher remains unchanged with full attention. All sparse distillation losses and hyperparameters are kept identical to full-attention distillation, as detailed in C.6. This simple substitution yields a better efficiency (50.9x acceleration) with no quality drop, as shown in Figure 3b Bottom.

## 3.4 Kernel Performance

As Figure 4b shows, VSA's fine block sparse kernel approaches the theoretical limit with nearly 7× speedup over FlashAttention-3 at long sequence lengths (85% MFU over FA3). Even after accounting for the coarse stage computations, VSA still maintains over 6× speedup. In contrast, FlexAttention [8] with an identical block-sparse mask (64×64 block size) achieves only a 2× speedup. Applying VSA's speedup to Wan-1.3B and Hunyuan brings 2-3×inference speedup, as shown in Figure 4a.

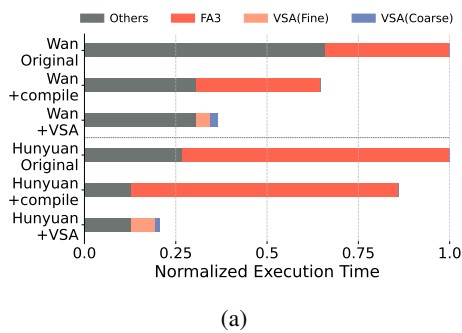
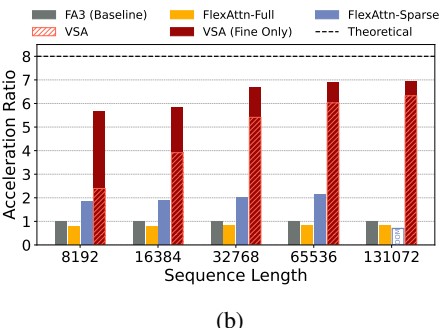

(a)

(b)

Figure 4: Kernel benchmarks. (a): Runtime breakdown of a single transformer block for Wan1.3B and Hunyuan. VSA reduces the attention latency by 6×. (b): Speed of VSA with a fixed 87.5% sparsity under various sequence length with head dim 64. VSA approach the theoretical 8× speedup over FA3.

## 3.5 Inspecting VSA

To gain deeper insight into VSA's mechanism, we inspect the block-sparse attention maps generated by the coarse stage of our finetuned 1.3B model. As illustrated in Figure 5(a-f), the predicted attention patterns are highly dynamic, confirming our hypothesis that effective sparse attention must be data-dependent rather than relying on predefined structures. Even within a single layer, different attention heads often exhibit markedly distinct behaviors. Many observed patterns echo established heuristics, such as local attention focused on tokens near the query (akin to sliding tile attention), or spatial-temporal attention concentrating on tokens within the same frame (d), the same temporal-width plane (e), or the temporal-height plane. Conversely, some patterns deviate from simple heuristics, displaying highly global characteristics (b) or a combination of local and global focus (c).

We quantify the accuracy of critical token prediction calculated as the sum of attention scores within the top-32 cubes selected by the coarse stage. As a baseline, a random selection of 32 cubes from the 386 total (for a (16, 28, 52) latent) captures only 8% of the attention score, as shown by the red plane in Figure 5(e). In stark contrast, VSA maintains a high accuracy rate, consistently achieving at least 60% in most layers and timesteps, and reaching as high as 90% in some instances. This underscores VSA's strong capability in identifying critical tokens. Critically, even if the fine stage misses a small portion of the attention weight, the direct output from the coarse stage can potentially compensate for this. Further examination of Figure 5 (e) reveals systematic variations in prediction accuracy. Accuracy tends to increase monotonically with the timestep. Across transformer layers, however, the accuracy rate exhibits a zig-zag pattern. These accuracy dynamics across layers and timesteps suggest avenues for future optimizations with adaptive Top-$\mathcal{K}$ value.

## 4 Related Work

**Sparse Attention in LLMs.** There has been a proliferation of fixed-pattern sparse attention mechanisms in large language models [3, 46, 1, 7, 13]. In practice, however, most LLM training (over 90% of total FLOPs) happens on short sequences (≤32K tokens) under the "train-short, adapt-long" paradigm [40, 23, 12, 42], so sparse attention saw little uptake beyond sliding window attention variants like in Mistral [17]. With LLMs targeting contexts beyond 1M tokens, sparse attention has seen renewed interest. Recent work primarily targets inference-time speedups [39, 49, 18, 41], while the latest methods explore trainable, dynamic sparsity patterns (MoBA [25], NSA [45]) to enable efficient end-to-end training on extreme-length sequences. We draw inspiration from them; However,

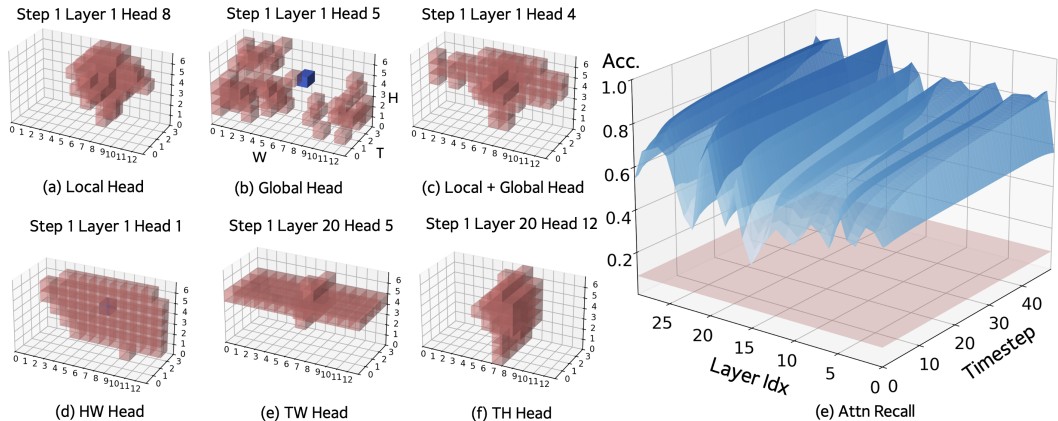

Figure 5: Visualization of the attention pattern of VSA. (a)-(f): VSA dynamically select different cubes to attend, where the blue cube indicates query and red cubes indicated selected key and values.(e): VSA critical-token prediction accuracy.

VSA differs from MoBA by directly contributing the coarse-grained attention output to the final representation and using smaller blocks compatible with efficient block-sparse kernels. Compared to NSA, the nature of video and bidirectional attention avoids grouped query constraints of the attention pattern. We discuss similarities and differences in depth in §E.

**Sparse Attention in Video DiTs.** Recent work has explored applying sparse attention post-hoc to DiTs pretrained with full attention [48, 6, 47, 37, 14] at inference. However, we argue that the case for trainable sparse attention in video DiTs is both distinct from and *more urgent* than in LLMs. First, video DiT demands far longer sequences, e.g., a 100K-token context yields only 5s video, making scaling inherently more costly than language models. Second, unlike LLMs, where long-context adaptation is a small fraction of total training, state-of-the-art video DiTs [20, 35, 29, 33] dedicate most of their compute budget to full-resolution, long-sequence training. As a result, these models remain bottlenecked by quadratic attention at both training and inference. This calls for trainable sparse attention mechanisms, like VSA, as a core design of video DiTs, not a post-hoc fix. DSV [33] also explores adding sparsity to attention during DiT training, but their multi-stage and profiler-based design may complicate the training pipeline, which we further discuss in §E.

## 5 Limitation and Conclusion

We present VSA, a trainable and hardware-efficient sparse attention tailored for scaling DiTs. Unlike prior work that applies sparsity post-hoc, VSA jointly learns to predict and apply attention sparsity at training and remains compatible with block-sparse compute layout. VSA currently operates with a fixed cube size of $(4, 4, 4)$, which requires video latent dimensions to be divisible by 4. While this may restrict the set of compatible resolutions, it can be addressed in practice by generating a slightly larger latent and cropping to the target shape. Another open question is how to determine the optimal sparsity level. While our scaling experiments (§3.2) provide preliminary insights, a complete understanding may require extending scaling laws to explicitly account for sparsity, alongside model size and training compute. Across diverse model sizes (60M to 1.4B) and budgets (up to $4 \times 10^{21}$ FLOPS), we show that VSA matches the performance of full attention at $2.53\times$ lower training cost, and achieves $85\%$ MFU of FA3. When integrated with Wan2.1-1.3B, it reduces end-to-end latency by $1.7\times$. We hope this work establishes trainable sparse attention as a practical and scalable alternative to full attention in further scaling video DiTs.

## Acknowledgments

We would like to thank Wei Zhou, Kevin Lin, Matthew Noto, Wenxuan Tan, and Jinzhe Pan for helpful discussion. The work is supported by UCSD HDSI, Nvidia, and a faculty research award from Google. The computing resources were provided by MBZUAI IFM and Nvidia's donation.

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

# A  Qualitative Examples

Prompt: An astronaut walking between stone buildings.

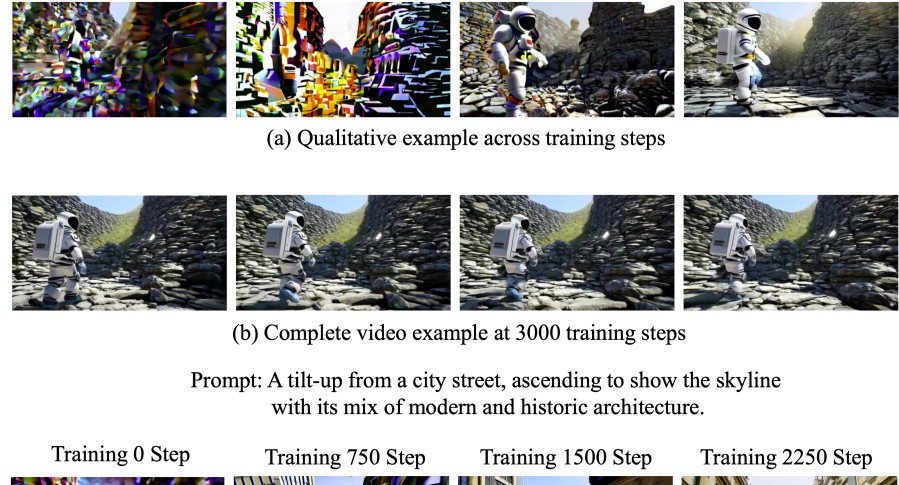

(a) Qualitative example across training steps

(b) Complete video example at 3000 training steps

Prompt: A tilt-up from a city street, ascending to show the skyline
with its mix of modern and historic architecture.

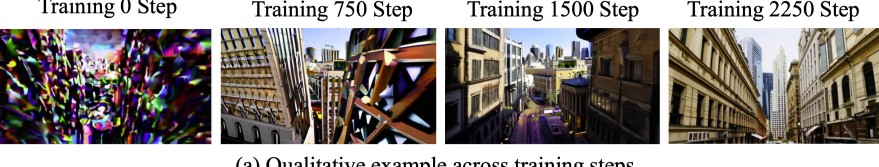

(a) Qualitative example across training steps

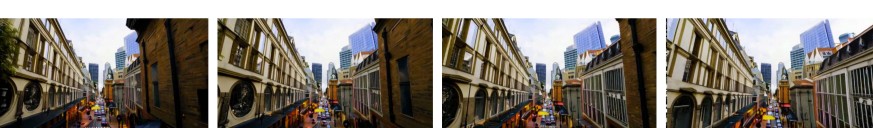

(b) Complete video example at 3000 training steps

Figure 6: Qualitative examples. In (a), we sample the same middle frame at each step of the video. In (b), we uniformly sample four frames across the video.

We qualitatively illustrate the finetuning process (§2.3§3) of Wan-1.3B in Figure 6. All frames are sampled from validation videos at selected training steps with $\mathcal{K} = 32$. At the start the training, the model exhibits noticeable artifacts when switching from full attention to VSA, reflecting the change in attention structure. As training progresses, the model gradually adapts to the sparse attention mechanism and recovers the ability to generate coherent videos.

# B  Pseudocode of VSA

We provide a pseudocode in a pytorch-like API for easier understanding of VSA.

```
def tile(x):
    return rearrange(x, "b h (n_t ts_t n_h ts_h n_w ts_w) d -> b h (
        n_t n_h n_w ts_t ts_h ts_w) d",
                     n_t=4, n_h=8, n_w=8, ts_t=4, ts_h=4, ts_w=4)

def untile(x):
    return rearrange(x, "b h (n_t n_h n_w ts_t ts_h ts_w) d -> b h (
        n_t ts_t n_h ts_h n_w ts_w) d",
                     n_t=4, n_h=8, n_w=8, ts_t=4, ts_h=4, ts_w=4)

q, k, v, gate = tile(q), tile(k), tile(v), tile(g)
coarse_attn_gate, fine_attn_gate = gate.chunk(2, dim=1)
```

```
# Coarse stage
B, H, L, D = q.shape
block = 64
topk = 32

q_c = q.view(B, H, L//block, block, D).mean(dim=3)
k_c = k.view(B, H, L//block, block, D).mean(dim=3)
v_c = v.view(B, H, L//block, block, D).mean(dim=3)

score = torch.matmul(q_c, k_c.transpose(-2, -1)) / (D ** 0.5)
score = torch.nn.functional.softmax(score, dim=-1)

output_coarse = torch.matmul(score, v_c)
output_coarse = output_coarse.view(B, H, L//block, 1, D).repeat(1, 1,
    1, block, 1).view(B, H, L, D)

# Keep only top-k blocks
topk_vals, topk_idx = score.topk(topk, dim=-1)
score = torch.zeros_like(score)
score.scatter_(-1, topk_idx, topk_vals)

score = score.view(B, H, L//block, L//block, 1, 1).repeat(1, 1, 1, 1,
    block, block)
score = score.permute(0,1,2,4,5,3).reshape(B, H, L, L)

# Fine stage
QK = torch.matmul(q, k.transpose(-2, -1)) / (D ** 0.5)
QK = QK.masked_fill(attn_mask == 0, float("-inf"))
QK = torch.nn.functional.softmax(QK, dim=-1)
output_fine = torch.matmul(QK, v)

# Combine stages with residual connection
hidden_states = output_coarse * coarse_attn_gate + output_fine *
    fine_attn_gate
hidden_states = untile(hidden_states).transpose(1, 2).flatten(2, 3)
```

Listing 1: Pseudocode of ViLAS in a pytorch-like API with no kernel optimization, assuming cube size (4,4,4) and video size (16,32,32). Note that the tile and untile operation can be moved to the beginning and end of the transformer to avoid calling them for each attention.

## C   Experimental Details

We document the detailed experiments setups for the results presented in Section 3.

### C.1   Model Architecture

We follow the architecture of Wan2.1 [35] for all experiments. Ablation studies are conducted on a 120M-parameter model initialized using the GPT-NeoX scheme, which leads to faster convergence than the default PyTorch initialization. The model adopts the pretrained UMT5-XXL [4] as the text encoder and the pretrained VAE from Wan2.1 for video tokenization. The architecture includes two types of attention: (1) self-attention among video tokens and (2) cross-attention for injecting textual information. Sparse attention is applied only to the self-attention layers. Detailed model configurations are provided in Table 2a.

### C.2   Ablation Experiments Setup

We train on long sequences of shape $61 \times 512 \times 512$, motivated by two factors. First, attention dominates the computational cost at this scale, making it the primary bottleneck. Second, sparse attention must be evaluated under long-context settings to demonstrate its effectiveness; short sequences do not present sufficient challenge.

Table 2: Model configuration and training hyperparameters used for the ablation studies.

| Model Config | Value |
|---|---|
| Head Dim | 64 |
| FFN Dim | 3072 |
| Cross Attn Norm | True |
| Freq Dim | 256 |
| In Channels | 16 |
| Out Channels | 16 |
| Num Heads | 12 |
| Num Layers | 12 |
| Patch Size | $1 \times 2 \times 2$ |
| QK Norm | RMS Norm (across heads) |
| Epsilon | $1 \times 10^{-6}$ |
| Text Dim | 4096 |

| Hyperparameter | Value |
|---|---|
| Learning Rate | $6 \times 10^{-4}$ |
| LR Scheduler | Constant |
| Warmup Steps | 100 |
| Batch Size | 1024 |
| Video Latent Shape | $16 \times 32 \times 32$ |
| Sequence Length | 16,384 |
| Attention FLOPs Ratio | – |
| Weight Decay | $1 \times 10^{-2}$ |
| AdamW Betas | $(0.9,\ 0.95)$ |
| Objective | Flow Matching [24, 22] |
| Timestep Sampler | LogitNormal$(0.0, 1.0)$ [9] |
| Total Traning FLOPS | $4.5 \times 10^{20}$ |

(a) 120M model used for ablation studies.   (b) Training hyperparameters

To establish a strong baseline, we perform a grid search over batch sizes $\{512,\ 1024,\ 2048\}$ and learning rates $\{5 \times 10^{-5},\ 1 \times 10^{-4},\ 2 \times 10^{-4},\ 6 \times 10^{-4}\}$. The best hyperparameters is used for all ablation variants. Training is conducted under a fixed compute budget of $4.5 \times 10^{20}$ FLOPs, which we find to be sufficient for training a 120M-parameter model – a 120M model with full attention outperforms a 60M model trained with $4 \times 10^{20}$ FLOPs, indicating FLOPS budget around $4.5 \times 10^{20}$ is compute-optimal for a 120M model. Each ablation job takes around 10 hours on 64 Nvidia H200 GPU. Full training hyperparameters are provided in Table 2b.

## C.3 Baseline Attention Variants

**Spatial-Temporal**   A widely adopted approach in early video generation works, including Open-Sora [50], OpenSora-Plan [21], LaVie [36], and Latte [27]. We alternate between spatial and temporal attention across layers.

**Spatial-Full**   In spatial-temporal attention, the temporal stage can become overly sparse. For example, with a latent shape of $(16, 32, 32)$, the temporal attention accounts for less than 1% of the FLOPs of full 3D attention. To mitigate this, we design a variant with four spatial layers and one full-attention layer every five layers.

**Compress KV**   This variant pools only the key and value tokens using a $2 \times 2 \times 2$ average pooling, reducing attention FLOPs by $8\times$. The query tokens remain at full resolution. This setup mimics the coarse-grained stage of VSA with a smaller pooling size and no pooling on query tokens.

**Strided Window**   Inspired by Swin Transformer, we propose a strided window attention that increases token interaction on top of spatial-temporal attention. Let $W_s$ and $W_t$ denote the spatial and temporal window sizes. For spatial attention, a query attends to all tokens in the same frame and to those in the same temporal window ($W_t = 2$). For temporal attention, a token attends to the same spatial location and to others in the same spatial window ($W_s = 8$).

**Conv Pooling**   Instead of mean pooling for block-level token aggregation, we use a 3D convolution with kernel size and stride of $(4, 4, 4)$ (same as the block size). The output channel dimension matches the head dimension.

## C.4 FLOPS Calculation

Elementwise operations such as LayerNorm and RoPE contribute negligibly to the total computational cost in transformers. Following the approximation in [15], we omit these operations and estimate the model FLOPs as $6ND$, where $N$ is the number of model parameters and $D$ is the number of input tokens.

However, for video DiTs trained on long sequences, attention computation becomes a dominant cost. We therefore incorporate the attention FLOPs following the formulation from FlashAttention [5]:

$$\text{FLOPs} = 6ND + 4 \cdot D \cdot S \cdot A \cdot H \cdot 3.5 \cdot L$$

where $D$ is the number of tokens, $S$ is the sequence length, $A$ is the number of attention heads, $H$ is the head dimension, and $L$ is the number of transformer layers. For sparse attention, we adjust the attention portion according to their sparse pattern.

### C.5 Sparse Adaptation Setup

To bridge the gap between full and sparse attention, we adopt a sparsity decay schedule that gradually reduces the number of cubes used in attention computation. The model is first trained with full attention for the initial 50 steps, to accommodate the changed resolution and aspect ratio. Thereafter, we decrease the number of attended cubes by 10 (i.e., reduce Top-$K$ by 4) every 50 steps, until reaching the target sparsity level (In our setting Top-$K = 32$). Unlike directly training the model with extremely sparse attention, our progressive decay schedule enables a smooth transition and mitigates training instability.

In the finetuning experiments for Wan-1.3B, we trained on 80,000 synthetically generated videos from Wan-14B, each with a resolution of $448 \times 832$ and 61 frames. To accelerate training and reduce memory usage, we preprocessed both the VAE latents and text encodings. Training was conducted on 32 H200 GPUs using DDP as the parallelism strategy. We set the per-GPU batch size to 1, applied a gradient accumulation of 2, and used a learning rate of $1e{-}5$. The training ran for 4,000 steps.

In the finetuning experiments for Wan-14B, we set the final sparsity to 0.9 and trained on 200,000 synthetic videos from Wan-14B, each with a resolution of $768 \times 1280$ and 77 frames. Training was conducted on 64 H200 GPUs using DDP as the parallelism strategy. We set the global batch size to 64, and learning rate to $1e{-}5$. The training ran for 4,000 steps.

### C.6 Sparse Distillation Setup

Our Sparse Distillation on Wan-1.3B uses 64 H200 with a per-GPU batch size of 1, and runs for 12 hours. We initialize the generator model with the pretrained weights of the base model and then replace the attention module with VSA (sparsity=0.8), while keeping real score and fake score models the same as the base model. All DMD-related hyperparameters are held fixed, including the number of denoising steps, the generator update ratio, and the guidance scale for real-score model. After 4,000 steps training, the 3-step generator achieves a 50.9× reduction in denoising time relative to the baseline model and can generate a 5-second video in approximately 5 seconds on a single H200.

## D  Coarse Stage Runtime

For shorter sequence lengths, the coarse stage over-head is more pronounced. Our profiling experiments using nsys (Table 3) reveal that Top-$\mathcal{K}$ selection dom-inates this runtime. Although we fused the kernels for attention scaling, softmax, and Top-$\mathcal{K}$ operations to reduce memory traffic and improve data locality, it only provided modest improvements. Since the coarse stage overhead becomes negligible at longer sequence lengths – our primary target – we did not pursue further optimizations in this work. However, coarse stage acceleration remains an important direc-tion for future research.

Table 3: Coarse stage runtime (μs).

| Breakdown | w/o fusion | w/ fusion |
|---|---|---|
| $\text{QK}^T$ | 0.046 | 0.046 |
| scale | 0.060 | |
| softmax | 0.095 | 0.912 |
| topk | 0.869 | |
| PV | 0.045 | 0.045 |

## E  Discussion and Extended Related Work

VSA builds upon insights from prior work on trainable sparse attention mechanisms in both language modeling [45, 25] and computer vision [51, 33]. This section situates VSA within this landscape, highlighting key similarities and differentiating design choices.

MoBA [25]: VSA shares conceptual similarities with MoBA, particularly in: (1) employing a coarse-grained stage that utilizes mean pooling, akin to MoBA's gating mechanism, and (2) using attention scores from this pooled representation to guide block selection for sparse attention. However, a key divergence lies in the utilization of the coarse-grained stage output. While MoBA employs pooled attention solely for block selection, VSA incorporates the output of its coarse-grained stage directly into the final attention output, potentially enriching global context representation. More critically, MoBA's implementation, which relies on token gathering and variable-length FlashAttention, constrains it to larger tile sizes (e.g., 512). This can limit the granularity of its sparsity patterns and reduce speedup efficacy, especially for sequences like those at 128K. In contrast, VSA is implemented with block-sparse attention leveraging smaller, fixed-size blocks (e.g., 64x64 as per our findings), aiming for a better balance between performance (Table 1d) and practical world-clock speedup (Section 3.1).

NSA [45]: The two-stage (coarse/compress and fine/select) architecture of VSA bears resemblance to NSA's design. However, fundamental differences arise from their target domains. NSA is tailored for causal language model decoding, where typically only a single query token is processed at a time. This necessitates specific strategies like group query attention to enable efficient kernel implementation(NSA can only pool Key-Value pairs). Video DiTs, operating bidirectionally on entire sequences, do not face the same single-query constraint, allowing VSA to apply pooling more broadly and employ distinct sparse patterns for different attention heads without resorting to grouped queries. Furthermore, NSA includes an additional sliding window stage for local information, a component VSA found unnecessary for video generation in our setup (Table 1b).

BiFormer [51]: Similar to BiFormer, VSA utilizes a coarse-grained, tile-to-tile attention mechanism. However, in BiFormer, this coarse attention serves only to derive the dynamic sparse pattern for the subsequent token-to-token attention and does not directly contribute to the final output. Our ablations (Table 1b) indicate that for VSA , the output of the coarse-grained stage is paramount for achieving optimal performance. Additionally, BiFormer's original implementation lacked direct FlashAttention compatibility, impacting its throughput compared to VSA 's design, which is optimized for hardware-aligned block-sparse operations.

DSV [33]: DSV represents pioneering work in exploring trainable sparse attention specifically for video DiT training. Both VSA and DSV aim to reduce the cost of identifying load-bearing regions. DSV achieves this by introducing dedicated low-rank attention predictors with a reduced head dimension, which are trained in a separate, multi-stage process and are not fully end-to-end integrated with the main DiT training. VSA , on the other hand, reduces this cost by performing attention on spatially pooled representations (e.g., from a 4x4x4 cube of tokens) within its coarse-grained stage. Crucially, VSA is designed to be end-to-end trainable, requiring minimal modifications to existing DiT training frameworks, unlike the more complex training system designed for DSV's predictors.

# F   Broader Impact

VSA aims to make high-quality video generation more accessible by significantly reducing the training and inference cost of video diffusion models. Our work may help democratize video creation tools for a broader range of users and use cases, including education, animation, and independent media production. However, as with many generative technologies, VSA also presents potential risks. In particular, the ability to generate realistic videos at scale may increase the risk of malicious applications, such as generating deepfakes or misleading media content. We emphasize the importance of developing robust detection tools, usage guidelines, and ethical standards in parallel with technical advances to ensure the responsible deployment of such models.

