# OpenReview forum: "Faster Video Diffusion with Trainable Sparse Attention"
_NeurIPS.cc/2025/Conference — NeurIPS 2025 poster_

### Official Review · Reviewer_dQZn · 2025-07-02

**Clarity:** 2
**Significance:** 2
**Originality:** 3
**Rating:** 4
**Confidence:** 4

**Summary:**

This paper introduces ViSA, a trainable, hardware-efficient sparse attention mechanism for video diffusion transformers. By using a two-stage coarse-to-fine block-sparse design aligned with GPU kernels, VISA achieves faster attention inference without sacrificing generation quality. Extensive experiments demonstrate the effectiveness of ViSA.

**Questions:**

Please see the weaknesses for the main questions. In addition, the anonymous github repo is empty.

**Ethical Concerns:**

["NO or VERY MINOR ethics concerns only"]

**Final Justification:**

My concerns are mostly addressed and I raise my rating to boarderline accept.

**Limitations:**

Yes

**Quality:**

2

**Strengths And Weaknesses:**

### **Strengths**
1. The trainable sparse attention is novel, which can eliminate train-test mismatch.
2. The block-sparse design is hardware-efficient and achieves real-world speedups compatible with Flash Attention.
3. Extensive experiments, including scaling laws and ablations, validate ViSA’s effectiveness across model sizes and training budgets.

### **Weaknesses**
1. The tile size and sparsity are predefined hyperparameters that require empirical tuning and may not generalize well to other resolutions or multi-scale training and inference.
2. Despite good quantitative results, the lack of qualitative visualizations makes it hard to judge the real impact of the sparse attention mechanism, especially given VBench’s limited coverage.
3. The VBench results reported seem inconsistent, with Wan’s baseline scores much worse than previously published (should be 84.92/80.10/83.96 for Quality/Semantic/Total).

---

> ### Author Rebuttal · Authors · 2025-07-30
>
> We thank reviewer dQZn for their  thoughtful feedback. We are glad that the reviewer appreciated our core contributions, including ViSA's novel trainable sparse attention mechanism, its hardware-efficient block-sparse design, and the comprehensive experimental validation. Below, we address the raised concerns regarding generalization, qualitative evaluation, and VBench reproducibility.
>
> > Tile Size Selection
>
> The tile size of 64 is indeed predefined. However, we believe this is one of the most important contributions of the paper. Through extensive ablation studies (Table 1(d)), we identify that a tile size of 64 achieves an great trade-off between attention granularity and hardware efficiency. **Rather than a limitation, we consider this fixed tile size a feature** that makes ViSA hardware-aligned and quality-optimized.
>
> > Sparsity Selection
>
> We acknowledge this as a limitation and explicitly discuss it in §3.2 and our Limitations section. Rigorously predicting the optimal sparsity level during pretraining likely requires an extension of the scaling law framework to include sparsity as an additional axis—a direction that is beyond the scope of our current training budget. That said, we empirically find that **sparsity levels between 0.8 and 0.9 consistently yield strong performance across a range of settings, including pretraining/finetuning/distillation, different model sizes, and various input resolutions**. This robustness is demonstrated both in our experiments in the paper and in the additional evaluations included below.
>
> > Generalization to Other Resolution
>
>
> Following your advice, we evaluate ViSA on two different configurations and report blind human preference scores (avoiding reliance on VBench) using 100 randomly sampled prompts from MovieGen-Bench:
>
> 1. ViSA finetuned on Wan2.1-14B-T2V, 77×768×1280 resolution, sparsity = 0.9:
>
>   |               | Full Attention Win | Tie | ViSA Win |
>   |---------------|--------------------|-----|----------|
>   | Hi-res (14B)   | 27                 | 47  | 26       |
>
> 2. ViSA on Wan2.1-1.3B-T2V with 3-step DMD distillation, 61×448×832 resolution, sparsity = 0.8:  We applied the DMD distillation recipe to Wan-1.3B-T2V using both full attention and ViSA (sparsity 0.8). For the ViSA variant, we replaced the student transformer's attention module with ViSA, leaving the teacher and critic components unchanged. After distillation, our model reduce inference diffusion steps from 50 to 3.
>
>
>   |               | Full Attention Win | Tie | ViSA Win |
>   |---------------|--------------------|-----|----------|
>   | Distilled (1.3B) | 23                 | 50  | 27       |
>
> The results suggest that ViSA generalizes well across architectures, input resolutions, and distillation settings. To the best of our knowledge, **ViSA is the first sparse attention mechanism to show strong performance under extremely low diffusion steps in DMD distillation**. In comparison, previous sparse attention methods were only tested with 50-step schedules, often retaining full attention in the early steps.
>
>
> > Visualization
>
> Our submission includes some visualization in Figure 7, Appendix F. While we are unable to include more videos/links/attachments in rebuttal due to this year’s NeurIPS policy, we directly addressed this concern through fully blind human preference studies presentend above. **Our fully-blind human evaluation confirms that ViSA-generated videos are visually stunning and indistinguishable from those generated with full attention**.  We will publicly release all code, checkpoints, and visualizations to support open evaluation beyond VBench.
>
> > VBench Scores and Reproducibility
>
>
> We would like to clarify that the VBench results in our submission were produced strictly following the VBench's official evaluation protocol, including the prompt provided by VBench.
>
> However, after going through VBench’s github issue, we discovered that the original Wan team applies an prompt rewriter not included in the official VBench repo’s guidelines, which substantially alters evaluation outcomes—particularly the Semantic score. Unfortunately, the Wan team has not released their rewritten prompts or their exact evaluation pipeline(random seed,guidance scale,scheduler shift...), which makes precise replication of their results impossible. We made our best effort to approximate their setup by rewriting VBench's prompts using Wan's official rewriter and re-evaluating both Full Attention and ViSA. The updated results are as follows:
>
>
> | Model      | Quality | Semantic | Total |
> |------------|---------|----------|-------|
> | Full Attn  | 83.71   | 77.98    | 82.56 |
> | ViSA       | 83.60   | 79.47    | 82.77 |
>
> These updated scores are now much closer to those reported for Wan. We emphasize that our evaluation has always been consistent and fair: ViSA and full attention were always evaluated by us under identical setups. While we cannot exactly match Wan’s numbers due to their unreleased setup and inherent randomness in generation, we will clearly document this caveat and provide full evaluation details in the final version. We also hope our additional results on human evaluation can complement the VBench scores.

---

> > ### Comment · Reviewer_dQZn · 2025-08-05
> >
> > Thanks the authors for the rebuttal. My concerns are mostly addressed and I increase my rating.

---

> > > ### Comment · Area_Chair_hkdr · 2025-08-06
> > >
> > > Dear Reviewer dQZn,
> > >
> > > Please do tell the authors any outstanding questions you might still have so that they may provide further response. Thank you very much.
> > >
> > > Your AC.

---

> ### Author Response · Authors · 2025-08-04
>
> We would like to thank reviewer  dQZn for the thoughtful feedback and valuable suggestions!  We are glad you found our approach to trainable sparse attention novel and the experimental evaluation thorough. We really believe your recommendations—such as testing across multiple resolutions and including benchmarks beyond VBench— has greatly strenghthen our paper! We will update all new results and necessary clarification into the camera-ready version.  If there are any remaining questions or areas that need clarification, we would be happy to address them.

---

### Official Review · Reviewer_WVmh · 2025-07-05

**Clarity:** 3
**Significance:** 3
**Originality:** 3
**Rating:** 4
**Confidence:** 4

**Summary:**

The paper "Faster Video Diffusion with Trainable Sparse Attention" introduces VISA (Video Diffusion with Trainable Sparse Attention), a novel method aimed at accelerating video diffusion models by reducing the computational cost of attention mechanisms. The key contribution is a trainable, hardware-efficient sparse attention mechanism that replaces full attention during both training and inference. VISA employs a hierarchical attention approach, consisting of a coarse stage that identifies critical tokens and a fine stage that computes token-level attention within these critical regions. The method is designed to be end-to-end trainable and compatible with modern GPU architectures, achieving significant speedups without compromising generation quality. The authors demonstrate that VISA reduces training FLOPS by 2.53× and inference time by 1.7× compared to full attention, while maintaining comparable performance.

**Questions:**

- Generalization to Other Datasets: How does VISA perform on other types of video data, such as higher-resolution videos or videos with different content characteristics? Can the authors provide additional experiments or analysis to demonstrate the generalizability of their method?

- Training Overhead: Can the authors provide more details on the computational overhead introduced by the coarse stage during training? How does this overhead compare to the overall computational savings achieved by VISA?

**Ethical Concerns:**

["NO or VERY MINOR ethics concerns only"]

**Final Justification:**

I keep my original score

**Limitations:**

The authors have adequately addressed the limitations of their work by discussing the current constraints, such as the fixed cube size and the dependency on specific hardware architectures. They also acknowledge the potential negative societal impact of their work, such as the risk of generating misleading or harmful content, and emphasize the need for robust detection tools and ethical guidelines. Constructive suggestions for improvement could include exploring methods to make VISA more adaptable to different hardware platforms and conducting additional experiments to validate its generalizability to diverse datasets.

**Quality:**

3

**Strengths And Weaknesses:**

Strengths:

- The paper is technically sound, with a well-supported methodology and comprehensive experiments. The authors provide extensive ablation studies and scaling-law experiments, demonstrating the effectiveness of VISA across different model sizes and training budgets. The results show significant improvements in computational efficiency without a drop in diffusion loss.

Weaknesses:
- Generalizability: While the experiments demonstrate impressive results on specific datasets and model sizes, it is unclear how well VISA generalizes to other types of video data or different diffusion model architectures. Additional experiments on diverse datasets would strengthen the paper's claims.
- Computational Overhead During Training: Although VISA reduces overall computational cost, the introduction of the coarse stage adds some overhead. The authors could provide more detailed analysis on the trade-offs between the benefits of sparse attention and the additional complexity introduced during training.

---

> ### Author Rebuttal · Authors · 2025-07-30
>
> We thank reviewer WVmh for highlighting the importance of evaluating generalizability and training overhead. Below, we address those concerns.
>
> > Generalization
>
>
> We conducted additional experiments to evaluate ViSA’s robustness across different models, resolutions, and training recipes. We then performed blind human evaluation over 100 prompts from MovieGen-Bench.
>
> - Low-step Setup via Distillation: We distilled Wan-1.3B-T2V from 50 diffusion steps to 3 steps with both full attention and ViSA (sparsity 0.8) using the DMD distillation recipe. For ViSA, we simply swapped the attention module in the student transformer, while keeping the teacher and critic unchanged.
>
>   |               | Full Attention Win | Tie | ViSA Win |
>   |---------------|--------------------|-----|----------|
>   | Distilled (1.3B) | 23                 | 50  | 27       |
>
> - High-resolution Finetuning: We fine-tuned Wan2.1-14B-T2V at resolution 77×768×1280 with ViSA (sparsity 0.9) and compared against full attention using the same evaluation protocol:
>
>   |               | Full Attention Win | Tie | ViSA Win |
>   |---------------|--------------------|-----|----------|
>   | Hi-res (14B)   | 27                 | 47  | 26       |
>
> These results show that ViSA achieves comparable generation quality to full attention across diverse model sizes, resolutions, and training recipes, despite computing only a fraction of the full attention cost. **We believe ViSA is the first sparse attention mechanism to demonstrate effectiveness under extremely low diffusion steps in DMD distillation**. In contrast, prior sparse attention methods were only evaluated with 50-step schedules, often keeping the first few steps as full attention.
>
>
> > Additional Complexity
>
> We clarify that ViSA **reduces**, rather than increases, training complexity:
>
> - In Figure 4b, we compare the runtime of ViSA (fine-only) and ViSA (coarse + fine). When the sequence length exceeds 30K (typical for SoTA video DiTs), the coarse stage overhead becomes negligible compared to the computational savings achieved by VISA. In Section 2.4, we mentioned that course stage's "runtime accounts for only 14%, even when the fine stage is 87.5% sparse."
> - Figure 2b shows that ViSA reduces training FLOPs by 2.53× while maintaining loss parity, consistently across scales.
> - From an engineering complexity perspective, ViSA is implemented as a standalone kernel package (like FlashAttention) and can be directly plugged into DiT codebases.

---

> > ### Author Response · Authors · 2025-08-04
> >
> > We would like to thank reviewer WVmh04  for the thoughtful and constructive review. We sincerely appreciate your suggestions, which helped us strengthen the paper—especially around generalizability and training overhead. We believe we have addressed your comments with new experiments and clarifications, and would be grateful if you could take a moment to review our responses. Please let us know if anything remains unclear—we would be happy to follow up.

---

> > ### Comment · Area_Chair_hkdr · 2025-08-06
> >
> > Dear Reviewer WVmh,
> >
> > Please kindly read the rebuttal posted by the authors and see if they have resolved all your questions. If yes,  please do tell them so. If no, please do tell them any outstanding questions you might still have. Thank you very much.
> >
> > Your AC.

---

### Official Review · Reviewer_oL6k · 2025-07-06

**Clarity:** 3
**Significance:** 3
**Originality:** 3
**Rating:** 5
**Confidence:** 3

**Summary:**

The paper introduces ViSA, a trainable sparse attention mechanism for video diffusion transformers (DiTs) that reduces the computational bottleneck of quadratic 3D attention. ViSA employs a two-stage approach to identify critical tokens and compute attention efficiently, achieving a 2.53× reduction in training FLOPS and a 6× speedup in attention time with comparable quality.

**Questions:**

NA

**Ethical Concerns:**

["NO or VERY MINOR ethics concerns only"]

**Final Justification:**

This paper extends the study of sparse attention to the training stage, achieving competitive performance while enabling VSA to adapt to various resolutions. Its effectiveness in video generation highlights broad practical value, and I therefore recommend acceptance.

**Limitations:**

yes

**Quality:**

3

**Strengths And Weaknesses:**

Strengths:

1. ViSA proposes a novel trainable sparse attention mechanism for video DiTs, learning critical token positions during training rather than sparsifying only at inference, thus better preserving model performance.

2. The design of ViSA fully takes into account the hardware characteristics of modern GPUs, ensuring compatibility with efficient attention implementations such as FlashAttention through block-sparse computational layouts, thereby translating theoretical computational savings into actual runtime acceleration.

3. The paper includes extensive ablation studies and scaling experiments, demonstrating ViSA's robustness across various configurations (e.g., tile sizes, sparsity levels) and model sizes.

4. This paper is well written and easy to follow.

Weakness: Although ViSA has limitations like fixed cube size and optimal sparsity determination, the authors provide practical solutions like cropping that still significantly enhance video DiT training and inference efficiency.

---

> ### Author Rebuttal · Authors · 2025-07-30
>
> We thank reviewer oL6k for the positive feedback and for highlighting the effectiveness of ViSA’s trainable sparse attention design.
>
> Regarding the noted limitation on fixed cube size and resolution compatibility, we have updated our implementation of ViSA to support **arbitrary video resolutions via a variable-size block kernel**. Specifically, when the input video shape is not divisible by the default cube size of (4, 4, 4), we now partition it into regular cubes wherever possible and pad the remaining non-full cubes to match the expected size. These padded cubes are processed normally in the coarse stage. For the fine stage, we improve our sparse attention kernel to explicitly mask out padded tokens, ensuring they never contribute to attention computation. This update removes the resolution constraint, enabling ViSA to support any video shape. The overall framework, including the differentiable coarse-to-fine attention and compatibility with block-sparse FlashAttention-style layouts, remains unchanged.

---

> > ### Comment · Reviewer_oL6k · 2025-08-04
> >
> > The author's response has addressed my concerns, and I will maintain my score as accept.

---

### Note · Authors · 2025-08-16

Dear Area Chair and Reviewers,

We would like to provide a short summary of the rebuttal discussion period as our final remark.

The initial reviews identified a number of points for clarification and further validation. These included:

- Fixed cube/tile size and sparsity tuning potentially impacts generalization.
- Resolution compatibility – whether ViSA works beyond specific input sizes.
- Generalizability – performance on other datasets, resolutions, and training recipes.
- Training overhead – whether the coarse stage increases complexity.
- Qualitative evaluation and VBench reproducibility – need for more human evaluation and clarity on baseline discrepancies.

In the rebuttal, we addressed each point with new experiments and clarifications:

- Introduced a **variable-size block kernel** to remove fixed-resolution constraints.
- Added **generalization experiments** across multiple model sizes, resolutions, and low-step DMD distillation settings, with blind human evaluation showing ViSA matches full attention in quality.
- Clarified that the  **coarse stage overhead is negligible** for typical sequence lengths.
- Explained **VBench discrepancies** and provided updated results plus human preference studies.

Following the rebuttal, reviewers responded positively:

- Reviewer oL6k confirmed their concerns were addressed and maintained their accept rating.
- Reviewer WVmh did not engage with us during the discussion period, but we believe all their concerns were addressed.
- Reviewer dQZn confirmed their concerns were mostly addressed and raised their rating.

In summary, the reviewers’ points were largely requests for clarification or additional evidence, and all have been addressed in full.

Thank you for your time and support throughout the review process!

Best regards,

ViSA Team

---

### Decision · Program_Chairs · 2025-09-17

**Decision:**

Accept (poster)

**Comment:**

This paper introduces a trainable sparse attention mechanism to accelerate video diffusion models. It adopts a two-stage approach to first identify critical tokens and then compute attention efficiently. The reviewers generally agreed that the proposed method is novel and hardware-efficient, and the evaluations are extensive. They raised some questions regarding generalizability, computational overhead, and lack of qualitative visualizations, which were addressed by the authors in the rebuttal and discussions. The final ratings are 1 "accept" and 2 "borderline accept". It is recommended to accept this paper for its novelty and efficiency.